# Situation of India in the COVID-19 Pandemic: India’s Initial Pandemic Experience

**DOI:** 10.3390/ijerph17238994

**Published:** 2020-12-02

**Authors:** Azizah F. Siddiqui, Manuel Wiederkehr, Liudmila Rozanova, Antoine Flahault

**Affiliations:** 1Institute of Global Health, University of Geneva, 1202 Geneva, Switzerland; liudmila.rozanova@unige.ch (L.R.); antoine.flahault@unige.ch (A.F.); 2Global Studies Institute, University of Geneva, 1205 Geneva, Switzerland

**Keywords:** COVID-19, India, epidemiology, non-pharmaceutical interventions, social disruption, economic impact, health systems, global health, case study, coronavirus

## Abstract

In this article, we investigate the impact of COVID-19 through screening and surveillance methods adopted in India, as well as the potential health system, social, political, and economic consequences. The research was done in a chronological manner, and data was collected between 30 January 2020 till 12 June 2020. Initial containment measures, including point of entry screenings and testing protocols, appeared insufficient. However, testing capacity was gradually expanded after the commencement of a nation-wide lockdown. Modeling predictions have shown varying results on the emergence of cases depending on the infectiousness of asymptomatic individuals, with a peak predicted in mid-July having over two million cases. The country also faces risks of the economic plunge by losing approximately 4% of its gross domestic product, due to containment measures and reduction in goods importation. The low public health expenditure combined with a lack of infrastructure and low fiscal response implies several challenges to scale up the COVID-19 response and management. Therefore, an emergency preparedness and response plan is essential to integrate into the health system of India.

## 1. Introduction

In December 2019, COVID-19 was first identified in Wuhan, China, as a respiratory tract infection causing symptoms, such as fever, chills, dry cough, fatigue, and shortness of breath [1]. This atypical viral pneumonia has disabled the world, causing catastrophic health and economic losses. The novel coronavirus belongs to the family of SARS and MERS-CoV, but the impact of the former is more crippling as illustrated by the exponential increase in infectious cases [2]. The incubation period of COVID-19 is between 1–14 days, a mean period of 6 days [1], during which asymptomatic carriers of the virus can transmit the disease to healthy people, as proven by the evidence of human-to-human transmission via droplets or contact [3]. COVID-19 was declared as a Public Health Emergency of International Concern by the end of January, according to the standards of International Health Regulations (2005) by the World Health Organization [4].

Due to the unprecedented spread of the virus, the world has gone into a virtual lockdown as several countries have initiated strict screening of potential cases introduced in their territory [5].

We investigate the medical, social, political, and economic impact of COVID-19 in India by conducting literature reviews, as well as sourcing information from articles, media reports, and other publicly available documents to contextualize relevant information. The study is a literature overview and part of a case series on various countries’ initial experiences on the COVID-19 pandemic. Data and literature were collected covering the period from the first case detected in India, 30 January 2020, until 12 June 2020. India is the country of interest for this research to examine the proposed containment measures and their efficacy, given the potentiality of an extensive transmission at the national-level in accordance with the current health management system and a population rivaling China. The study will inspect the current health system capacity of India, the epidemiological situation, and how screening and surveillance of patients can immobilize the spread of the contagion. The study will be conducted chronologically, according to the timeline following the first COVID-19 case introduced in India to the subsequent transmission of the disease throughout the country.

## 2. Case Presentation

### 2.1. Characteristics of the Country

The Indian peninsula is bounded by the Arabian sea, the Bay of Bengal, the Indian Ocean to the south, and the Himalaya, which separates India from the Asian mainland. India’s predominant climatic conditions are influenced by the Himalaya, and the monsoons create a tropical climate with hot and humid temperatures in summer [6].

India’s population count exceeds 1.3 billion people and is further growing. With a registered birth rate of 20.2 per 1000 people and a death rate of 6.3 per 1000 people, India has a positive growth rate. More than half of India’s population is under age 30. The life expectancy at birth increased over the past decades, and as of 2016 is 67.4 years for men and 70.2 years for women. India’s infant mortality rate is 33 per 1000 live births in 2017; rural areas report higher mortality rates than urban areas [6].

India is politically organized as a federal republic that consists of several states and union territories (UT). The country is governed in a multi-party parliamentary democracy that operates on a constitution adopted in 1949. Through its federal power division between union and state, governments on a local level have the autonomy to legislate concerning law and public health [7]. The country’s constitution recognizes 22 languages for state-official correspondence—a snippet of India’s multi-ethnicity. Almost 80% of the Indian population are Hindu. Other religious groups are Muslim (~14%), Christian (2.3%), Sikh (1.7%), and others/unspecified (2%) [7]. On the other hand, according to the Sustainable Development Goals (SDG) India Index Report 2019, India’s diversity brings many forms of disparities, such as: “[…] inequalities in income and consumption; structural inequalities based on gender, religion, caste and social groups as well as regional inequalities, all of which manifest in inequalities of opportunities and access” [8] (p. 131).

With a growth rate of 6.8% (2018–2019), India is one of the fastest-growing economies in the world. The Indian GDP is estimated to be US$2.72 trillion, with a per capita income of US$2015 in 2018. Regarding poverty reduction, people living below the World Bank’s International Poverty Line dropped from 21.2% in 2011 to 13.4% in 2015. Despite the economic growth, India’s unemployment rate is still above the global average and differs between rural (5.3%) and urban (7.8%) areas [8,9]. According to the Human Development Report 2019, India reaches an HDI score value of 0.647, which puts the country in the medium human development category, ranking 129th out of 189 countries [8].

### 2.2. India’s Health Care System

The Indian healthcare system is divided between the Union government and State governments, according to the federal system [10]. The Union Ministry of Health and Family Welfare heads the programs to be implemented, which can be eventually adopted by the state government, while the state government overlooks the public health system within the state. A national-level health quality and control are done jointly by the union and state governments [10]. The health care infrastructure is divided into primary, secondary, and tertiary level-health centers. Primary-level health center includes sub-centers for marginalized populations, rural, as well as urban regions for disease prevention and health promotion [10]. National Health Policy 2017 has made primary health centers a major focus, committing 2/3rd of its resources for building and maintaining the centers as the first point of contact with individuals [11]. Secondary-level health center includes community health centers for medical specialties, surgeries, and radiological equipment. Tertiary-level health centers comprise of university and district hospitals for specialized medical care. Private health facilities of India are generally secondary, and tertiary level centers [10]. The health care system of India officially includes Ayurveda, Yoga and Naturopathy, Unani, Siddha, and Homeopathy, collectively known as AYUSH, with a dedicated ministerial office called Ministry of AYUSH [12]. AYUSH deals with herbal, ergonomic, and traditional medicine-based treatments for ailments and certified practitioners of AYUSH are recognized as authorized healthcare professionals [13].

India received an overall rating of “CCC” and was placed the furthest amongst the nine countries weighted in the Laura Miller ranking system, presenting a health system needing definite improvements in its capacity [14]. Moreover, the Indian government spends a meager 3.5% of its total gross domestic product (GDP) on health almost consistently since 2006. This percentage is approximately half of the overall world GDP spent on health systems by WHO member states, as well as the average current health expenditure on health by BRICS nation, both standing at 6.3%. (cf. Appendix A) [15]. However, the National Health Profile by the Central Bureau of Health Intelligence reported that the government’s public expenditure (GPE) for health is just 1.28% of the total government revenue, indicating that private health expenditure and out-of-pocket payment (OOP) is very high [6].

OOP is an insistent issue in India as nearly 65% of the total health expenditure according to the National Sample Survey Office, Ministry of Statistic and Program Implementation 2013–2014 [16]. Universal Health Coverage Global Monitoring Report by WHO [17] summarized that Indians were spending nearly 80% of their household income on medications. A long period of hospitalization results in borrowing assets and/or utilizing income savings for treatment costs, where catastrophic healthcare leads to the impoverishment of the patients. To tackle this issue, Ayushman Bharat Pradhan Mantri Jan Arogya Yojana (AB PM-JAY) was launched in 2018, covering 40% of the poorest population in India and is amongst the biggest health care insurance system in the world, aiming to reach universal health coverage [18]. Lack of health care providers is a major concern as there are 35 doctors, nurses, and midwives per 10,000 population, and one allopathic public doctor per 10,000 individuals [6,19]. Patients are more likely to visit private health facilities than governmental health facilities, due to superior health provision and quality in private providers. Moreover, private health facilities establish a monopoly in rural regions where individuals pay more for services than urban regions [20]. Gender parity is still significant as male patients pay approximately 100 dollars more than female patients, due to demographic, socioeconomic, and discriminatory conditions [21]. As per OECD data, India has a modest 0.5 bed per 1000 habitants, compared to China’s 4.3 beds per 1000 habitants [22].

Indian Council of Medical Research (ICMR) is at the forefront of medical research in India, and research for COVID-19 is primarily coordinated by it. ICMR is looking over test-kit production and purchase, pharmaceutical usage, daily case count, and guidelines for tackling the virus [23]. All the stages of research and guidelines have been transparent and timely updated. The current method of COVID-19 testing is by RT-PCR nasal and/or throat swab, with the testing probes procured from the USA by ICMR-NIV. ICMR has also validated non-US FDA EUA/CE IVD kits for quick commercial use in India [24]. Any suspected COVID-19 case should be notified to the district surveillance unit by registered physicians, medical officers, as well as AYUSH practitioners. Rapid antibody kits for COVID-19, however, tests positive for SARS-CoV-2 infection, not specifically COVID-19. Although the COVID-19 test can cost around INR 4500, ICMR has appealed to make the tests free or heavily subsidized, especially for private laboratories [25,26].

### 2.3. Epidemiological Situation of COVID-19 in India

The following section presents the epidemiological situation regarding COVID-19 in India based on data retrieved from the Johns Hopkins University Centre for Systems Science and Engineering (JHU CSSE), World Health Organization (WHO), and Government of India (GoI). However, there are minimal deviations in epidemiological data between the three references.

On 30 January 2020, India reported the country’s first case of COVID-19 in Kerala. The index case was a student returning from Wuhan and was isolated in a hospital. As of 3 February, a total of three cases were confirmed in Kerala, with all initial cases coming from different cities. By 20 February, they were declared recovered [27]. Little information was provided regarding the initial COVID-19 cases in India, and thus, it is unknown whether they were contacts of the first case or whether they had travel history. However, after a month lag the number of cases started to surge, affecting more states and union territories by the beginning of March (c.f. Appendix B). According to the Ministry of Health and Family Welfare, the transmission of COVID-19 is mainly related to travel and local transmission of imported cases, limited community transmission was first reported on 30 March [28]. On the contrary, Klein et al. assume that community transmission in India most likely started at the beginning of March [29].

On 14 March, India reported its first two COVID-19 related deaths. Accordingly, both patients were of age >65 years and with comorbidities [27]. Throughout the first weeks after the outbreak onset until mid-May, India’s case-fatality ration (CFR) remained stable at a constant 3.2%. As of 9 June, CFR dropped to 2.8% by 0.6 deaths per 100,000 capita. India’s CFR resembles the aggregated CFR from the South East Asia Region [27].

The availability of segregated data was spare. However, according to the press release by the Ministry of Health and Family Welfare on 6 April, 76% of confirmed cases were male. The age distribution of the confirmed cases presented, as following—47% below 40 years, 34% between 40 to 60 years, and 19% were 60 years and older. Furthermore, segregated mortality data was reported: 73% of reported deaths were male (27% female). Although only 19% of cases are among elderly people, 63% of reported deaths account for the age group 60 and above. 30% are among the group between 40 to 60 years, and 7% were younger than 40 years. Moreover, 86% of the deceased suffered from comorbidities [30].

Prior to the date of submission, India counted more than 300,000 cases, recording a relatively constant increase of daily incidences (cf. Figure 1). This means that India did not yet manage to substantially contain the spread of COVID-19. The doubling time of case-counts steadily decreased currently to eight days (9 June 2020).

## 3. Management and Outcome

### 3.1. Mathematical Projections of COVID-19 in India

Ray et al. operated an eSIR-Model to predict future case counts in India. In addition to the three components susceptible, infected, and removed (recovered or deaths) of SIR-Models, the authors incorporated time-varying transmission rates [31]. Hence, the transmission rates accounted for a non-intervention period (b = 1), a period with a travel ban (0.8), another period with a travel ban and social quarantine (0.6), and lastly, a period with a nationwide lockdown (0.2). R_0_ was defined as 2. The projection is based on observed data up to 18 March 2020. The authors predicted for each scenario cumulative case-counts on given dates ranging between 1 and 161 million on 15 May 2020 (cf. Figure 2). The case-counts of this scenario are based on the total population of the cities Mumbai, Pune, Kochi, and Bengaluru located in current COVID-19 hotspots [31].

Based on an SEIR-Model, Mandal et al. analyzed the effectiveness of point-of-entry screening of symptomatic and asymptomatic travelers at an airport [32]. Derived from a simulated epidemic in Wuhan, the authors estimated the number of COVID-19 cases being introduced into the community in India, accounting for cases being detected prior to leaving China. The model projection for the time to reach 1000 cases in India ranged from 45 to 47.5 days when screening for symptomatic travelers only. Additionally detecting 90% of all asymptomatic cases would delay the average time to the epidemic by 20 days. Hence, the authors question the impact of point-of-entry screening as it is infeasible to reach such coverage [32].

A second model projected the mitigation of COVID-19 in the four most populated metropolitan areas Delhi, Mumbai, Kolkata, and Bengaluru. Based on the assumption that 50% of symptomatic cases would be identified and quarantined (R_0_ = 1.5), the cumulative incidence and peak prevalence could be reduced by 62 and 89 per cent, respectively. As a result, the outbreak is prolonged by factor 3 compared to the scenario without quarantine coverage. However, in this scenario, asymptomatic were not considered infectious. In a pessimistic scenario where R_0_ = 4 and asymptomatic cases are infectious, cumulative incidence and peak prevalence are only reduced by two and eight per cent. In this case, the projection of the epidemic duration (time over which the prevalence of symptomatic COVID-19 infections is >1 case) was only approximately 1.5 times longer (ca. 550 days) than without quarantine intervention [32].

Ambikapathy and Krishnamurthy modeled the impact of various lockdown scenarios and predicted future cases to a period of 110 days, starting from 2 February 2020 (lockdown was implemented on day 45) [33]. Lockdown periods of 14, 21, and 42 days significantly reduced the number of cases on day 110 from 378,036 to 42,950 cases. However, augmented transmissibility was modeled to assess the effect of panic shopping or mass travel prior to implementing the lockdown. These enhanced exposures could mitigate the effect of a lockdown [33].

Chatterjee, K et al. projected case-counts based on an SEIR model, including epidemiological data from March 2020 [34]. Accordingly, the model predicted 2,979,928 cases by 25 May 2020 with peak prevalence by mid-July. Implementing quarantine measures could reduce cases by 90%. This results in a total of 241,974 cases by 25 May under the condition of 50% quarantine compliance of people with COVID-19 [34].

Klein et al. computed a mathematical model using IndiaSim, a model that encompasses information on population demographics, socio-economics, location, and access to healthcare [29]. The baseline scenario without interventions foresees between 300 to 400 million cases with a peak-prevalence of 100 million cases in April/May. The optimistic scenario, a trajectory, including lockdowns with full compliance, decreased virulence, and temperature/humidity sensitivity, reduced peak load by approximately 75%. In this scenario, the peak is projected to be around mid-June [29].

Both Ray et al. and Klein et al. discuss the effect of humidity and temperature on the spread of COVID-19 [29,31]. Ray et al. found a negative correlation between temperature and COVID-19 case counts. However, these results were not statistically significant. Klein et al. estimated R_0_ to be between 2 and 3 and reduced it to 1.8 when adjusting for seasonality [29]. Likewise, they notice that evidence on the effect magnitude of humidity and temperature is not sufficiently described [29].

It is unclear whether COVID-19 will be extinct or reoccur in annual or seasonal cycles. To predict and prepare for future emergences of COVID-19, ICMR collects extensive data from bats to detect potential spillover events [35].

Kumar and Roy used the Bailey’s Model to assess the removal rate of infectious to recovered individuals. When the number of infected equals the number of recovered, the transmission of the virus has mostly stopped, and the epidemic is considered extinguished [36]. Through regression analysis, the authors projected that the coefficient will reach 100% in mid-September. At this point, this includes epidemiological data of infectious and removed individuals [36]. As of 9 June 2020, the coefficient reached more than 48% and approximately aligned with the predicted removal rate.

### 3.2. Screening and Testing

Through rigorous point of entry surveillance efforts, by the end of March, India conducted thermal scan on more than 1.5 million passengers at the airports, placing thousands of passengers under surveillance in home isolation [27]. As of 5 February, in addition to the National Institute of Virology in Pune, India increased testing capacity with additional eleven laboratories [27]. Until 9 March, India further increased the network of laboratories fit for testing COVID-19 to 52 and expanded the network to more than 100 laboratories by the end of March [27]. The number of tested samples increased accordingly. At the beginning of February, the laboratories tested 49 samples. This number first only moderately increased to 2880 by 28 February before testing accelerated to 22,928 samples by 25 March 2020 [37]. By mid-April, India further increased testing capacity to 229 private and governmental laboratories testing more than 15,000 samples summing up to a total of 195,748 samples tested [27,38,39]. On 18 May, India reached a landmark by performing 100,000 tests in one day [27]. As of 12 June 2020, India ramped up its capacity to a total of 885 laboratories fit for testing COVID-19, conducting more than 125,000 tests a day, resulting in a total of 4,666,386 samples tested [27,40]. However, despite the increased capacity from 130 tests per million capita on 11 April to currently 3780 tests (status 11 June 2020), India conducts remarkably fewer tests compared to other countries [41]. Ray et al. doubt that the current cases reported reflect the actual epidemiological situation and mention that “the number of truly affected cases, which depends on the extent of testing, the accuracy of the test results and in particular frequency and scale of testing of asymptomatic cases who may have been exposed“ [31] (pp. 2–3). They assume that either there is an underreporting of cases, due to the difficulties of getting tested or that India’s strict measures (e.g., rigorous point of entry screenings) indeed managed to contain the outbreak successfully [31].

To contain the spread of the disease, ICMR reviews and updates the testing strategy periodically. Accordingly, ICMR defined criteria for testing asymptomatic patients and patients with influenza-like illnesses on 20 March and 9 April, respectively [42,43]. ICMR conducts an IgG ELISA test sero-survey to estimate and monitor the pandemic within specific population groups. The first results show that 0.73% of the enrolled 26,400 individuals were contracted with SARS-CoV-2 leaving a large proportion of the population still susceptible. However, the risk of infection is 1.09 and 1.89 times higher in urban areas and slums, respectively [44].

Generally, studies modeling the projection of case-counts in India tend to overestimate the epidemiological magnitude in comparison to the officially reported numbers. Nevertheless, the current epidemiological development resembles (at lover counts) the scenario modeled by Rey et al., with a lower transmission rate through travel ban and social distancing [31]. In the optimistic scenario, Klein et al. projected the peak prevalence by mid-June [29]. However, given the current daily incidence trends (mid-June), India might not have reached peak prevalence, yet. Furthermore, the deviation of modeled projections and the current development of the outbreak might be explained by the low test capacity, and therefore, the Indian Government has been criticized for reporting suspiciously suppressed data [45]. According to Narayanan et al., extensive testing is needed to better target public health interventions. Pooling testing samples can detect COVID-19 prevalence cost-effectively, and thus, is suitable in resource-limited environments [46].

### 3.3. Travel Restrictions

After the announcement of COVID-19 as a public health emergency of international concern (PHEIC) on 30 January 2020 [4], the civil aviation authority began universal health screening of international passengers at the entry point from China by temperature check and filing self-declaration forms [27]. Entry screening expanded for other nations as the virus began spreading globally, in line with the International Health Regulations 2005 for point of entry screening [47]. States with no international airports and/or seaport began monitoring the influx of travelers from rail and road to check for potential cases [27]. On 11 March, when WHO declared COVID-19 as a pandemic, Indian authorities eventually banned visas and non-essential travels from affected countries, listed as China, Iran, Italy, South Korea, France, Spain, and Germany. Although the International Health Regulations advises against reduced mobility in terms of travel [47], travel restrictions began on 13 March as visas issuance was restricted to essential travel and delegates only. Subsequently, all international passengers entering India were required to go through screening tests. The travel ban expanded to all European countries and nations of the Middle East on 18 March [48].

Passengers with COVID-19, arriving from affected countries, were put in quarantine for 14 days in the port of arrival city, while asymptomatic and/or healthy passengers were advised to commence home quarantine, testing for COVID-19 should symptoms appear [49]. Passengers’ left hand was stamped with inedible ink to maintain the date and time for home quarantine, a move that could risk assault, due to stigma towards COVID-19 suspects [50]. Individuals violating the quarantine stage could be penalized with Indian penal code section 188, 269, and 270, that is, “violation of order promulgated”, “negligently doing any act known to be likely to spread infection of any disease dangerous to life” and “malignantly doing any act known to be likely to spread infection of any disease dangerous to life”, respectively [51].

On 25 March 2020, the government of India imposed a sudden complete nationwide lockdown for 21 days, with the closure of non-essential markets and a complete halt to all national rail network, international and domestic flights [27]. However, the restriction turned into a challenge for daily wage workers and migrant laborers who cannot continue to earn their living nor return to their hometowns, due to the closure of rail and road networks [52]. The short-notice before commencing lockdown stranding thousands of migrants garnered severe criticism as the government had not managed the migrant crisis despite releasing 20 lakh crore rupees (265 billion dollars) relief funds to tackle COVID-19, resulting in scores of migrants walking back home for miles, essentially risking viral transmission through their long journey [52]. The home ministry has urged to create of temporary shelters for citizens affected by the restriction, with all states and UTs expected to follow suit [53]. However, the mass exodus of migrant laborers continued to occur, as only limited dedicated buses and trains for such migrants were arranged by the state government. Moreover, due to approaching summer and the heat waves, many migrant laborers perished in their journey home [54].

India went under four phases of lockdown extensions and entered its fifth phase on 8 June, where regions deemed safe, called “green zones,” will have more liberty in movements and business operations, whereas danger “red” zones will continue strict travel and trade restrictions [53]. However, limited domestic air travel and railway travel with appropriate safety precautions for citizens in necessity resumed on 25 May and 1 June 2020, respectively. An “unlock” phase coincided with the 5th lockdown to restart selected businesses, educational institutions, and local public transport, while maintaining distance and hygiene [27,53].

### 3.4. Health Recommendations

On 30 January 2020, with the advice of WHO, the government of India initiated awareness of proper hygiene and sanitation steps to protect from the spread of diseases [48]. A major focus was put on proper handwashing, covering oneself, while coughing and sneezes, social distancing, thorough cooking of meat and dairy, and avoiding contact from wild or farm animals. WHO country office of India (WCO) worked with ICMR and National Centre for Disease Control for building laboratory capacity and disease surveillance, respectively [48].

By 9 March 2020, WCO, along with the Ministry of Information and Broadcast, directed all telecom operators of India to launch a special COVID-19 caller tune to raise awareness about the prevention strategies as cases began to rapidly rise after Italy’s surge of cases [27].

In a bid to amplify sales, Reckitt Benckiser, prominent health, hygiene, and home products company, released a liquid handwash advertisement, vilifying rivals Hindustan Unilever’s soap bar by claiming that the formers liquid handwash is more effective for cleaning hands. Bombay High Court eventually suspended the ad for one month from 22 March to 21 April to stop unverified claims and to malign soap bars [55]. By 21 March 2020, the Ministry of Consumer Affairs, Food and Public Distribution Department of Food and Public Distribution has increased the line of ethanol/Ethyl Alcohol/Extra Neutral Alcohol (ENA) production by states and union territories of India to meet the urgent requirements by health facilities [56].

After the imposition of the first lockdown at the end of March, the government released several guidelines on protection methods, such as making face masks compulsory in public places, but preferably using reusable masks, leaving medical masks for health professionals [57]. Public guidelines, such as social distancing, avoiding spitting in public, and avoiding mass gathering was enforced. Following mass misinformation spread among the citizens through social media, regarding false remedies and fake news, the government of India launched a ‘MyGov Corona Helpdesk’ on the highly popular social media application WhatsApp, to receive Indian-centric accurate and verified information [57]. The rumors were also addressed by WHO and physicians regarding false claims of alcohol and garlic intake as a cure, along with universal mask usage and panic purchase of goods to be avoided [57].

Dedicated Health Facility (DHF) follows the triage system of segregating patients according to the severity of disease symptoms in a temporary emergency ward in tertiary facilities with premium amenities [58]. The patients are put in separate units based on the confirmed cases of COVID-19 and suspected cases awaiting test results and/or persons who were in close contact with a confirmed case and are treated accordingly. Adhering to this system, while following proper safety measures, for both health workers and patients, may prove to be an effective response technique for outbreak management [58].

### 3.5. Financial Expenditure

By the middle of March 2020, State Disaster Response Fund (SDRF), under the Disaster Management Act, has recognized COVID-19 as a national calamity, releasing 25% of its funds for clothing, shelter, alimentation, health screening, and contact tracing with the National Health Mission [59]. SDRF also has given a 10% fund allowance for building lab capacity and strengthening surveillance. The Global Fund has also assisted India by providing up to 5% of its grant to fight against COVID-19 [60]. In a bid to produce a COVID-19 vaccine and medicines at low cost, the Prime Minister had pledged 15 million dollars to global health partnership for immunization, GAVI, the international vaccine alliance [27].

### 3.6. Economic Impact

The impact of COVID-19 on the Indian economy is still unfolding. Measures to mitigate the spread of COVID-19 (e.g., 21-day lockdown) will pressure down India’s growth projections [61]. Barclays estimates that the cumulative costs of the nationwide shutdown will be approximately $120 billion or 4 percent of India’s gross domestic product (GDP). Several forecasts reduce GDP growth projections for 2020 by 1.7 to 3.1 percentage points [62]. Moody’s [61] expect a grow rate of 2.5% for 2020 followed by 5.8% in 2021 (cf. Figure 3a). Moreover, Singh et al. [63] expect a downward trajectory of inflation (CPI and WPI) as the oil price falls and productivity and demand decrease. Despite the pressure on currency fluctuations and price level stability, India is expected to benefit from the tumbling oil prices. India covers 80% of its oil demand with foreign sources, and is thus, a significant petroleum importer. Ultimately, India’s gain from low oil prices might be moderate, as it also reduces tax revenues and stimulates currency depreciation relative to US$ [63].

The United Nations Conference on Trade and Development, UNCTAD, [65] estimated India’s trade impact to be US$ 348 million caused by manufacturing slowdown in China and the following disrupted global trade. Divided by sector, the projected impact is most for the chemical sector (US$129 million), followed by textile and apparel at US$64 million and the automotive industry at US$34 million (cf. Figure 3b).

More than 80 percent of India’s workforce earn their money in the informal sector and are particularly affected by the lockdown [50]. The INR 1.7 lakh crore (US$ > 22.5 billion) relief package under Pradhan Mantri Garib Kalyan Yojana provided food security measures and direct cash transfer. The volume of the package amounted to approximately 1% of the country’s GDP, which aims to help the poorest segment of the population [66].

However, according to Singh et al. [63], quantifying the magnitude of the economic impact is difficult as it depends on the development of the COVID-19 outbreak and the proactive measures taken by the Government. Given India’s limited fiscal room, economic stimulus measures increase the country’s debt level, and thus, might slower pace of recovery [63].

### 3.7. Impact on Indian Pharmaceutical Industry

With Chinese production activities suspended, Indian pharma companies are threatened by goods in short supply. China delivers almost 70 percent of the active pharmaceutical ingredients (API) for medicines produced by Indian companies, leaving them vulnerable in maintaining its supply chain [67]. In addition, hoarding purchases created an artificial shortage of API, leading to a bulge in the price for paracetamol, vitamins, and penicillin [67]. At the same time, as a protective measure, the GoI installed an export ban on essential medicines [68]. Both disruption in supply and export restrictions threaten the availability of essential medicines and generics, especially “in the context of the COVID-19 pandemic, global reliance on Indian generics is likely to become a complex international challenge” [69] (p. 3). Considering the production capacities of Indian pharmaceutical companies, preventing impairments of their production and supply chains will increase the preparedness for large scale production for COVID-19 diagnostic tools and potential vaccines. Consequentially, this will not only support India’s economy, but also contribute to the global response in tackling this outbreak [69]. There is a political will to incentivize the industry to increase domestic API manufacturing capacity to decrease dependence on Chinese imports and strengthen national security [67].

Newton et al. [70] describe the risks to the supply and quality of tests, drugs, and vaccines imposed by insufficient evidence regarding COVID-19 and shortages that thrive substandard or falsified drugs. For India, the Medicine Quality Monitoring Globe Index reported issues related to substandard or falsified medical products. There were reports on fake vaccines or hand sanitizers sold at exorbitant prices [71]. Anti-malarial drug hydroxychloroquine was substantially used as the prophylaxis to COVID-19 [72]. India initially banned the export of hydroxychloroquine from meeting the domestic demands; however, the ban was partially uplifted after the US government requested the export of hydroxychloroquine for virus prevention. The drug was to be exported to 20 more countries placing requests for the tablets [73]. India supplies 70% of the world’s hydroxychloroquine and is aiming to export 250 million hydroxychloroquine tables to countries seeking medicine [74]. However, due to a lack of evidence on the efficacy of the drug [75], WHO recommended hydroxychloroquine to go under solidarity trial in over 35 countries [76].

### 3.8. Social and Political Disruption

As the stringent lockdown began in Wuhan, Indian air fleet Air India evacuated over 700 Indian and foreign nationals stranded in the city by carrying multiple batches of flight [77]. Air India also repatriated Indian crew and passengers trapped in the ill-fated Diamond Princess cruise ship. Indian Air Force evacuated 112 nationals stranded in Wuhan, 76 Indians, and 36 foreign nationals to Delhi, while also providing 15 tonnes of medical equipment and safety kits to China [77]. Following the subsequent surge of COVID-19 cases in Italy, commercial airlines Air India evacuated Indians from Rome and Milan, as well as Iran. All evacuees were taken to quarantine in Delhi [78].

On 14 March 2020, all public gathering areas, such as cinemas, malls, marriage halls, pubs, marathons, and night-fests, were closed [27]. Section 144 of the Indian penal code on unlawful assembly by more than four people was imposed to avoid gatherings [30]. The Ministry of Home Affairs has postponed indefinitely the decennial 16th National Population Census for 2021, originally to be conducted during April [53]. The Prime Minister of India announced 22 March 2020 as ‘Janta Curfew’ to solicit social distancing, before enforcing a nationwide lockdown for 21 days, starting 25 March [53]. Any person caught violating the lockdown for non-essential reasons were baton charged by the police deployed in the city streets. However, baton charge has received criticisms, as reported by Human Rights Watch [50], due to excessive force used by the police. To combat the financial and social burden of the underprivileged citizens of India stranded, due to the lockdown, the finance minister announced a substantial relief package under Pradhan Mantri Garib Kalyan Yojana for a period of 3 months on 26 March 2020. Under the scheme, eligible people will receive either monetary or alimentary assistance according to their specific needs [66].

COVID-19 updates have been available online through a crowdsourced website counting COVID-19 cases in India, launched by a group of volunteers to give an estimate of cases occurring daily all over the country by gathering data from state press releases, official government links, and reputed news source [79]. Another website sources its data from the Ministry of Health and Family Welfare [80] to present explicit, graphical data of distribution of infectious cases.

Xenophobic racism has risen towards the North-eastern citizens of India, possessing Mongoloid facial features, by mainlanders as per reports by Rights and Risk Analysis Group (RRAG) [81]. With no defiant law against racism in the country, perpetrators are penalized with the nearest substitutional law on attacks against women, but no such law exists for men. Lack of anti-racism law thereof has posed risks of verbal and physical abuse towards the ethnic groups [81]. WHO cautioned member states against stigmatization and discrimination in accordance with Article 3 of IHR (2005) during its second meeting by IHR (2005) Emergency Committee for Novel Coronavirus outbreak [35].

The government of India launched a mobile application called Aarogya Setu on 2 April 2020, for citizens to be informed about their potential risk of infection, medical advisories, and health practices to contain COVID-19, to self-assess their symptoms, as well as contact tracing. The application is mandatory for domestic travelers to download onto their smartphones to assist in contact tracing [27].

While the Indian government was praised for imposing an early lockdown, India’s most prominent epidemiologist was against the idea of lockdown, due to a lack of civil organization [82]. Amid the government’s controversial CAA bill, passed at the end of 2019, aiming at providing Indian citizenship to people of certain religious backgrounds, civil unrest arose before the pandemic hit the country, stigmatization persisting as Muslims were initially blamed for the spread of infection in India [83]. The offensive behavior expanded to healthcare workers who were dispelled as “carriers of the infection” and denied entry to their own homes by neighbors. Moreover, the Prime Minister’s suggestion of “self-reliance” during the pandemic raised questions, due to the country’s scarce resources unable to meet demands for healthcare provision, exhibiting complacency in adopting risk management strategies [83].

## 4. Discussion

The spread of COVID-19 rooting from China in December 2019 to a global scale has been categorized as a pandemic by the World Health Organization on 11 March 2020 [84]. COVID-19’s asymptomatic transmission has turned into a challenge to trace the exact source of viral spread. India’s rigorous point-of-entry screening may be infeasible in delaying the epidemic outbreak as the authorities conducted only thermal scanning for symptomatic passengers, thereby disregarding asymptomatic passengers as disease carriers.

India’s healthcare system has limited capacities and a strong focus on primary health care delivery. The country’s healthcare expenditure is 3.5% of the national GDP. However, only 1.28% of government public expenditure of the total government revenue is used for health care expenditure, indicating a high OOP burden.

The country’s limited health infrastructure capacities, as indicated in this paper, might result in higher case-fatalities. According to Khan et al., when adjusted for healthcare expenditure, existing burden of non-communicable disease, the demographic profile of the country and population density, capacity for the health care negatively correlated with case-fatalities [85]. There are further implications of limited health care capacities, including the provision of primary health services. As demonstrated by Garg et al., infrastructural limitations not only compromised patient safety and infection control measures, it further weakened the provision of outpatient services, particularly for services related to maternal and child health [86].

After detecting the first case on 30 January 2020, India experienced a delayed growth in the case-count. The initial spread was mainly driven by imported cases and contained local transmissions. However, there are indices that community transmissions prevailed by March 2020. Subsequently, India recorded a constantly increasing daily incidence rate. As of 9 June, Indi’s cumulative prevalence exceeded 300,000 COVID-19 cases with a doubling dime of eight days. A skewed age-sex distribution is apparent, with the majority of cases being men and under 40 years of age.

As of 9 June, the country reported a CFR of 2.8%, with the majority of deaths accounting for men. Segregated by age, the majority of fatalities occurred among people aged older than 60 years. However, notably, 37% of the reported deaths account for people younger than 60 years of age.

Several mathematical projections modeled the outbreak according to different scenarios accounting for quarantine measures. Depending on the scenario, efforts to contain the spread of the epidemic can reduce peak prevalence severely and extend the duration of the outbreak consequently. However, early mathematical projection of the spread of COVID-19 in India appeared to overestimate the trajectory when compared with the reported outbreak development. The effectiveness in point-of-entry screening was questioned by showing the limited impact on delaying the spread of COVID-19. Furthermore, in consideration of augmented transmissibility prior to implementing a nation-wide lockdown through panic shopping or mass travel might mitigate the effect of such a containment strategy.

India’s initial surveillance response was mainly based on thermal point-of-entry screening. As indicated, the point-of-entry screening strategy by thermal scanning of only symptomatic passengers may be inefficient. This is supported by the findings of Quilty et al., in their baseline scenario, showing an estimate of 46% of travelers with COVID-19 would not be detected. The authors conclude that airport screening is unlikely to detect a sufficient proportion of contracted travelers to prevent local transmissions from imported cases [87].

Furthermore, despite the continuously increased laboratory capacities, India conducts fewer tests compared to other peers. The large difference in modeled trajectories and current case-counts might indicate the magnitude of undetected cases. The assertion of having high estimated numbers of unknown cases is coherent regarding the relatively few tests conducted. As pointed out by Narayanan et al., pooled testing might capture the magnitude of the outbreak more accurately [46]. In a theoretical model calculated by Guha, Guha, and Bandyopadhyay, pooled testing reduces screening time and costs. It significantly reduces the proportion of misclassifications of those tested positive. Lastly, their results show that pooled testing is preferable for prevalence estimation to individual testing [88].

In March, India implemented visa restrictions for countries with a high COVID-19 burden and advised home quarantine for asymptomatic travelers entering India. On 25 March, the Indian government imposed a complete 21-day lockdown, including the suspension of domestic travel, closure of recreational places, gathering restrictions, and closure of non-essential businesses. In addition, the government released a relief fund with a volume of 20 lakh crore rupees (265 billion dollars). The government started deconfining by easing domestic travel restrictions on 25 May and implementing a deconfinement strategy with several lockdown phases accounting for regional epidemiological differences. Along with the WHO’s social media awareness campaign via WhatsApp, India also released an Indian-centric version of the COVID-19 campaign to provide evidence-based information and curb the regulation of myths and false news and contact tracing smartphone application mandatory for domestic travelers.

The magnitude of the economic impact is still unfolding as cases are increasing, and the government is compelled to take drastic measures for the management of COVID-19. Forecasts predict a plunge in GDP growth for India to the extent of 3.3% compared to estimation prior to the COVID-19 outbreak. As a low-middle income country, the challenge of fiscal responsibility on the growing demand of medical supplies added to the already low government public expenditure on health is a major concern not to be overlooked as it might slow the recovery rate of the Indian economy. The global slowdown in supply and production might have far-reaching consequences for the Indian pharmaceutical industry, notably an essential driver of the Indian economy. Moreover, Indian pharmaceutical companies may have significant capacities to produce preventive or therapeutic medical products crucial to the global response in tackling COVID-19.

This article provides insight into the initial pandemic situation in India. However, this paper is methodically limited as it is rather a literature overview than a systematic review. Additionally, to estimate the magnitude of underestimation of COVID-19 incidence and prevalence, further epidemiological indicators (e.g., test positivity rates) could have been included. However, neither WHO nor the Ministry of Health and Family Welfare further provided more in-depth data.

## 5. Conclusions

After detecting the first case on 30 January 2020, India experienced a delayed growth in the case-count. However, India’s low testing efforts might imply an understatement of COVID-19 cases as testing strategy and capacities were slowly expanded. Moreover, the overtly selective strategy of only screening symptomatic passengers also underrepresented the actual case counts. This further emphasized the presence of unreported cases in India. A universal testing strategy for all symptomatic, asymptomatic, presymptomatic, and paucisymptomatic cases is necessary to adequately mitigate the growing trend of COVID-19 spread, especially in India given the massive population of the country and increased risk of community transmission.

India’s healthcare system maybe be dented if there is a superfluous hospital admission, due to lack of adequate infrastructure and medical experts in relation to the high number of potential patients needing intensive care, given the already low expenditure on the public health system, standing at 1.28% of the total governmental revenue. Moreover, the catastrophic cost of testing and treatment for patients who are ineligible for insurances and government subsidies will further deepen debt and poverty in the country.

Mathematical models have presented both an optimistic and pessimistic scenario in regard to the lockdown and incoming passengers screening system. A stringent quarantine measure was unanimously observed to be the most efficient method to control the spread of the virus, however, extending the disease prevalence. However, models tend to overestimate the development of case prevalence. Moreover, the meteorological phenomenon has shown an inefficient effect on the incidence of COVID-19. The assertion of having high estimated numbers of unknown cases is coherent regarding the relatively few tests conducted by the Indian government.

Risk communication and community engagement is a necessary strategy to reduce fake news and propaganda for tackling COVID-19, which the government of India has proactively following, given the widespread dissemination of misinformation via social media platforms in India.

The pandemic of COVID-19 is likely to cause an economic crisis in India as approximately 4% of the GDP is projected to be lost amid the management and recovery phase. Moreover, as a low-middle income country, India is also moderately dependent on donations and funding by international organizations to control the spread of the disease. Further, the continuous loss of jobs and efflux of migrant laborers post lockdown phase reflected the lack of civilian employment sustainability by the government. This shows that a proper emergency and preparedness response plan is essential to avoid catastrophic loss in the financial sector and the already deprived health sector, which India must integrate into its core public health program.

With India’s 1.3 billion population and limited expenditure on public health, and the various predictions of COVID-19 development, the impact of the disease on India’s health infrastructure and economy may be difficult to explicate with accurate information, but it may likely cause a dent in the current health and financial system.

## Figures and Tables

**Figure 1 ijerph-17-08994-f001:**
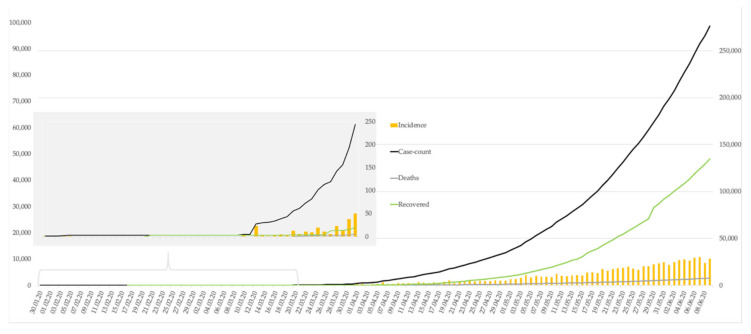
Epidemiological progress of the COVID-19 outbreak in India. Incidence is scaled according to the left axis. Data were derived from Johns Hopkins Center for Systems Science and Engineering (JHU CSSE).

**Figure 2 ijerph-17-08994-f002:**
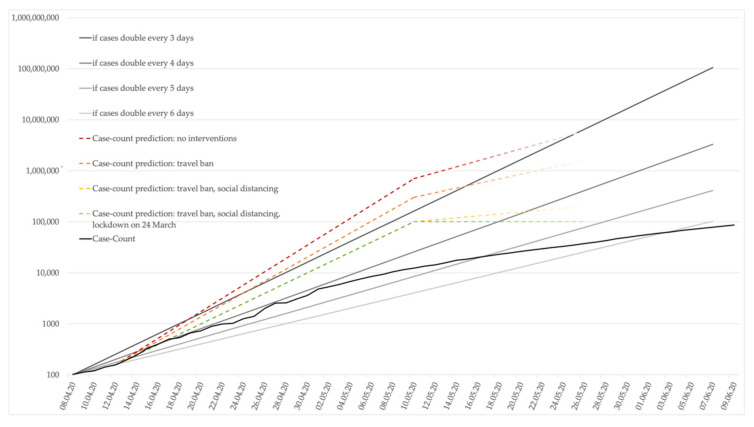
Logarithmic display of COVID-19 case-count and prediction, according to Ray et al. [31] (until 15 May) in relation to doubling time from the 100th case onwards.

**Figure 3 ijerph-17-08994-f003:**
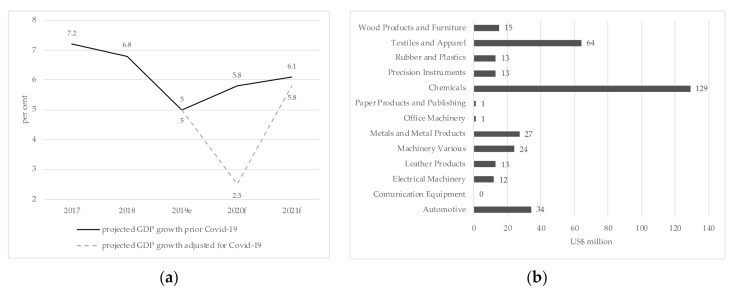
(**a**) Projected COVID-19 impact on GDP growth according to Moody’s. Data on reference GDP growth estimations prior to COVID-19 were gathered from World Bank [64]; (**b**) effects of China’s slowdown on India’s industry. Data shows the impact in US$ from a 2 percent reduction of Chinese exports in intermediate inputs [65].

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
