# Peer review of "Situation of India in the COVID-19 Pandemic: India’s Initial Pandemic Experience"

_ijerph, 2020, doi:10.3390/ijerph17238994_

Round 1
Reviewer 1 Report
This article reviews the current pandemic situation in India. Unfortunately, the text of the article contains a high level of plagiarism. Also, I'm afraid this is not a case report, as indicated in the section title and article title.
This article is neither a controlled study nor an observational study.
Presumably, this is a review of the current situation with the epidemic based on various publications.
Article structure:
1. Introduction
2. Case Presentation
2.1. Characteristics of the country
2.2. India’s health care system
2.3. Epidemiological situation of COVID-19 in India
3. Management and Outcome
3.1. Mathematical projections of COVID-19 in India
3.2. Screening and Testing
3.3. Travel restrictions
3.4. Health promotion
3.5. Financial expenditure
3.6. Economic impact
3.7. Impact on Indian pharmaceutical industry
3.8. Social and political disruption
4. Conclusion and discussion
For a review article, the content of the sections should be changed.
p.4 l. 152
Appendix 2
There is no Appendix 2 in the article.
Please check it.
p.4 l. 152, l.308
UT
Please decipher this abbreviation.
p.6 l. 260
influence-like illnesses
Perhaps the authors mean "Influenza-like illness"
p.7 l. 273
suspicious data [42].
Please specify what is "suspicious" about that data?
p.7 l. 285
Despite against
Please check it
p.9 l. 382
US$ 34
US$ 34 million
p.9 l. 391
dept level
Do you mean "debt?" Please check it
p.9 l. 401
[65] (p. 3)
[65, p. 3]
p.15 l. 634
42. Prof. Hanke, S.,
Please do not use "professor" in the reference section.
Author Response
Thank you for the honest and critical revision of our article. We tried to thoroughly improve our article accordingly. Major changes can be found in the abstract (to comply with not having more than 200 words), introduction, Epidemiological situation of COVID-19 in India (more in-depth of initial cases), Discussion and Conclusion (both separated to have an improved structure of both, the overall article as well as the mentioned chapters. Lastly, we revised our citations and referencing.
This article reviews the current pandemic situation in India. Unfortunately, the text of the article contains a high level of plagiarism. Also, I'm afraid this is not a case report, as indicated in the section title and article title.
This article is neither a controlled study nor an observational study.
Presumably, this is a review of the current situation with the epidemic based on various publications.
Article structure:
1. Introduction
2. Case Presentation
2.1. Characteristics of the country
2.2. India’s health care system
2.3. Epidemiological situation of COVID-19 in India
3. Management and Outcome
3.1. Mathematical projections of COVID-19 in India
3.2. Screening and Testing
3.3. Travel restrictions
3.4. Health promotion
3.5. Financial expenditure
3.6. Economic impact
3.7. Impact on Indian pharmaceutical industry
3.8. Social and political disruption
4. Conclusion and discussion
For a review article, the content of the sections should be changed.
This article is part of a case series reporting on the initial outbreak experiences of various countries. Thus, we changed it to case study as this is in accordance with the other articles of this case series. Likewise, the content and structure of this case series was provided by the University . However, we separated Discussion and conclusion in other to better incorporate the reviewer's feedback.
p.4 l. 152
Appendix 2
There is no Appendix 2 in the article.
Please check it.
p. 4, l. 210
Appendix 2 was changed to Appendix B
p.4 l. 152, l.308
UT
Please decipher this abbreviation.
p. 4, l 210
union territories was deciphered
p.6 l. 260
influence-like illnesses
Perhaps the authors mean "Influenza-like illness"
p. 6 l. 371
changed to "Influenza-like illness"
p.7 l. 273
suspicious data [42].
Please specify what is "suspicious" about that data?
p. 7 l. 387
changed to "suspiciously suppressed data" (in the sense of underreported data)
Note: we have changed the reference here to not include social media references. It has been replaced with the the news article.
p.7 l. 285
Despite against
Please check it
p. 7, l. 399
Sentence was reworded.
p.9 l. 382
US$ 34
US$ 34 million
p. 9, l. 512
Corrected to US$ 34 million
p.9 l. 391
dept level
Do you mean "debt?" Please check it
p. 9, l. 521
Change to "the country’s debt level"
p.9 l. 401
[65] (p. 3)
[65, p. 3]
p. 10, l. 536
[70] (p. 3), format is according to the journal's template (former [65] (p. 3))
p.15 l. 634
42. Prof. Hanke, S.,
Please do not use "professor" in the reference section
p. 16, l. 955
The references Prof. Hanke, S. was referenced according to the social media reference style, which includes the name of the social media account. Hence, Prof. appeared. However, as stated above, this reference was replaced by Sharma, A., Coronavirus numbers in India are low — but so is the testing rate, in DW. 2020, Deutsche Welle. in order to remove social media references.
Reviewer 2 Report
Situation of India in the COVID-19 Pandemic: A Case-Report of India’s initial Pandemic Experience
Abstract
The present research is a case-study aimed at investigating India’s efforts in trying to limit the spreading of COVID-19 through screening and surveillance methods and the healthcare, social, political and economic consequences faced by the country from the first reported cases to the period when the article was submitted.
The present abstract appears to be incomplete. More specifically, the authors should explain the main conclusions of their study. Moreover, it is made up of 263 words, thus not respecting the guidelines of the present journal, requiring an abstract length of maximum 200 words.
Finally, on line 24, the acronym GDP should be defined in brackets, as it is the first time it is mentioned. Since other such mistakes can be found elsewhere in the text (see, for example, page 1, lines 37-38 with the acronym “COVID-19” or page 2, line 76 with “SDG”), the authors should carefully revise all acronyms to make sure they are written in full the first time they are mentioned in the text.
Introduction
The introduction seems to be well-written and it is in line with the journal guidelines, stating that case reports should include “a succinct introduction”.
Some minor changes could be applied:
- page 1, lines 37-38 (“During December 2019, COVID-19 had originated from a seafood market in Wuhan, China, a respiratory tract infection causing symptoms such as fever, chills, dry cough, fatigue and shortness of breath”.): since there are no certain details about where Covid-19 originated, it would be more appropriate to omit that it originated in a seafood market.
- pages 1-2, lines 44-46 (“Covid-19 was declared as a Public Health Emergency of International Concern by the end of January, according to the standards of International Health Regulations (2005) by the World Health Organization.”): the date “2005” is not relevant and should therefore be removed.
- page 2, lines 47-48 (“Due to the unprecedented spread the virus, the world has gone into a virtual lockdown as several countries have initiated strict screening of potential cases introduced in their territory.”): the opening part of this sentence should be changed to “Due to the unprecedented spread of the virus”.
- page 2, line 51 (“The authors of this case study will investigate the medical, social, political and economic impact of COVID-19 in India by descriptive observational study from the first case detected in India until the date of submission of the article.”): since the submission date is not reported in this article, it would be helpful to explicitly indicate it or to indicate the time length – possibly expressed in months - of the data considered.
Case presentation
The present section provides an overview of the situation in India, starting from a more general description of India to then focus on India’s health care system. A more in-depth analysis of the case would probably be appropriate.
Some minor changes should be made, listed as follows:
- page 2, lines 72-74 (“Through its federal power division between union and state, governments on a local level have the autonomy to legislate, for example law and public health”): in order to make the sentence clearer and therefore easier to understand, the expression “for example” could be substituted with, for instance, “concerning”.
- page 2, lines 75-76 (“Other religious groups are 75 Muslim (~ 14%), Christian, Sikh, Buddhists, and Jains”): in order to be politically correct, if the percentage for Muslims in India is to be reported, the percentages for the other religious groups should be also added.
- page 3, line 138 (“ICRM has also validated non-US FDA EUA/CE IVD kits for quick commercial use in India”): the acronym “ICMR” was here misspelt to “ICRM”.
- page 4, lines 152-154 (“On 14 March, WHO [25] provided details of the first two events of death, both patients were of age >65 years and with co-morbidities. According to the press release by the Ministry of Health and Family Welfare 76% of confirmed cases were male”): a formula should be found to connect the first sentence with the second one. Moreover, since this paper aims at in-depth analyzing the cases in India from the first ones to the submission date of the article, the authors could expand on the analysis of the first reported cases.
- page 4, line 167 (“The doubling time of case-counts steadily decreased t currently 8 days (9 June 2020).”): the authors mistakenly typed a “t” in this sentence.
Measurement and Outcome
This paragraph offers some interesting insights especially concerning mathematical projections of COVID-19 in India, however, the following list reports some suggested changes to be applied to the present section:
- page 5, line 183 (“The case-counts of this scenario are based on the total population of the cities Mumbai, Pune, Kochi and Bengaluru located in current COVID-10 hotspots [28].”): the term “COVID-10” should be substituted with “COVID-19”.
- page 5, line 205 (“However, an augmented transmissibility was modelled to assess the effect of panic shopping or mass travel prior the implementation of the lockdown could mitigate its effect [30].”): this sentence is articulated in such a way that it is difficult to understand, it could therefore be expressed differently, as to make it easier and more immediate to understand.
- page 5, line 208 (“Chatterjee, K et al. [31]”): the initial of the first author’s name (“K”) should be omitted as – according to the journal guidelines - references in the text should only include surnames.
- page 6, lines 237-239 (“Through rigorous point of entry surveillance efforts, by the end of March, India temperature screened more than 1.5 million passengers at the airports, placing thousands of passengers under surveillance in home isolation.”): the expression “temperature screened” is not immediate to understand and could thus be substituted with a another, possibly clearer, expression.
- page 7, lines 280-281 (“Entry screening expanded for other nations as the virus began spreading globally, in line with the International Health Regulations 2005 for point of entry screening”): the date “2005” should be removed.
- page 7, line 285 (“Despite against the international health regulations 2005 [44]”): the sentence seems to be built in an incorrect way, the authors are therefore suggested to rephrase it.
-page 7, lines 292-294 (“Passengers’ left hand was stamped with inedible ink to maintain the date and time for home quarantine, a move that could risk assault due to stigma towards COVID-19 suspects (Human Rights Watch, 2020)”): the reported reference does not follow the journal guidelines, stating that references should be numbered in order of appearance and “In the text, reference numbers should be placed in square brackets [ ]”).
- page 8, line 318 (“3.4. Health promotion”): a possible substituting title, which would seem more appropriate for the content of the present section, could be “health recommendations” or, alternatively, “health guidelines”.
- page 8, lines 336-343 (“After the imposition of the first lockdown at the end of March, the government released several guidelines on protection methods such as making face masks compulsory in public places, but preferably using reusable mask, leaving medical masks for health professionals. Public etiquettes such as social distancing, avoiding spitting in public and avoiding mass gathering was enforced. Myths surrounding coronavirus arose regarding false remedies, creating panic among the citizens. Following mass misinformation spread among the citizens through social media, the government of India launched a ‘MyGov Corona Helpdesk’ on the highly popular social media application WhatsApp, to receive Indian-centric accurate and verified information.”): no references were reported here; the authors are suggested to add some references, thus allowing the reader to possibly expand their knowledge by consulting the reported sources.
- page 8, line 338 (“Public etiquettes such as social distancing, avoiding spitting in public and avoiding mass gathering was enforced.”): the expression “public etiquette” could here be substituted with a more appropriate one as, for instance, “health recommendations”.
- page 9, line 368 (“Moreover, Singh et al. [58] expect a downward trajectories of inflation”): the term “trajectories” should be made singular “trajectory”.
- page 9, lines 386-387 (“it is aimed to help the poorest of the poor [62].”): the expression “the poorest of the poor” is not adequate for the scientific language used in articles and could therefore be changed.
- page 10, lines 408-410 (“Newton et al. [66] describes the risks to the supply and quality of tests, drugs and vaccines imposed by insufficient evidence regarding COVID-19 and shortages that thrive substandard or falsified drugs.”): since “Newton et al.” is a plural subject, the verb should also be plural and, therefore, “describe” should be written.
Conclusion and discussion
Firstly, the title of the present section should be totally rewritten. The the part of the discussion must be extended in an appropriate way, integrating literature that better justifies the results achieved. It should also be clarified in what terms the results presented are original, explaining why for India the situation can be considered particular. The part of the conclusions must be written separately, clearly summarizing the main result obtained from this survey. In addition, the limits and future developments should be highlighted more clearly in the final part.
The journal guidelines explicitly state that the article should include “a conclusion briefly outlining the take-home message and the lessons learned”, the authors could therefore clearly define the take-home message for this study.
Further possible changes are listed as follows:
- page 11, lines 469-470 (“The spread of COVID-19 rooting from China in December 2019 to a global scale has been categorized as a pandemic by the World Health Organization on 11 March 2020”): a specific reference from the World Health Organization could be added here.
- page 11, line 472 (“India’s rigorous point-of-entry screening might be of infeasible in order to delay epidemic outbreak.”): this sentence could be re-phrased as to make it clearer and therefore easier to understand.
- page 11, line 478 (“India’s healthcare system maybe be dented if there is a superfluous hospital admission due to 477 lack to infrastructure and medical experts.”): the expression “lack to” should be changed to “lack of “.
References
Concerning the referencing, the authors seem to respect the journal guidelines as for in-text citations (“In the text, reference numbers should be placed in square brackets [ ], and placed before the punctuation; for example [1], [1–3] or [1,3].”). Conversely, the authors do not seem to follow the journal guidelines for references at the end of the text, therefore, major changes should be applied to this section. Moreover, the authors do not include the DOI (digital object identifier) for articles, as explicitly suggested in the aforementioned guidelines.
One last consideration concerning the references should be made about sources. The authors should possibly include as much scientific literature available. Other, non-scientific, literature (such as magazines) can also be relevant in this article but they should avoid citing, for example, social media contents, such as tweets (see reference number 42).
Use of English
English language is used properly in the present article. Although sometimes the way sentences are built is not fluid, most of the times the message the authors want to express is clear and immediate to understand.
Overview
In general, the main strength of the present research is the detailed description of a global issue with interesting insights upon the pandemic situation in India. An important weakness of the present case report is the fact that the authors completely overlook to describe the first reported cases of COVID-19 in India as indicated by authors’ main objectives.
However, the present case report requires moderate changes both at the level of the formatting requirements and of its content - especially the “Case Presentation” and “Conclusions” sections - in order for it to be adequate for publication.
Author Response
Thank you for the detailed feedback and constructive improvements. We have made the following changes and/or corrections as suggested by the reviewer.
Abstract
The present research is a case-study aimed at investigating India’s efforts in trying to limit the spreading of COVID-19 through screening and surveillance methods and the healthcare, social, political and economic consequences faced by the country from the first reported cases to the period when the article was submitted.
The present abstract appears to be incomplete. More specifically, the authors should explain the main conclusions of their study. Moreover, it is made up of 263 words, thus not respecting the guidelines of the present journal, requiring an abstract length of maximum 200 words.
The abstract has been rewritten to be under 200 words, giving a better explanation of the study with a concluding phrase, see line 13-27 , page 1.
Finally, on line 24, the acronym GDP should be defined in brackets, as it is the first time it is mentioned. Since other such mistakes can be found elsewhere in the text (see, for example, page 1, lines 37-38 with the acronym “COVID-19” or page 2, line 76 with “SDG”), the authors should carefully revise all acronyms to make sure they are written in full the first time they are mentioned in the text
As COVID-19 is a name on its own for the disease, we have left it as it is . example line 13 page 1.
Introduction
- page 1, lines 37-38 (“During December 2019, COVID-19 had originated from a seafood market in Wuhan, China, a respiratory tract infection causing symptoms such as fever, chills, dry cough, fatigue and shortness of breath”.): since there are no certain details about where Covid-19 originated, it would be more appropriate to omit that it originated in a seafood market.
Line 32, page 1- Sentence has been rephrased from (COVID-19 originated form a seafood market in Wuhan) to "COVID-19 was first detected in Wuhan")
- pages 1-2, lines 44-46 (“Covid-19 was declared as a Public Health Emergency of International Concern by the end of January, according to the standards of International Health Regulations (2005) by the World Health Organization.”): the date “2005” is not relevant and should therefore be removed.
Line 39-41, page 1- we mentioned the year 2005 as it was the last time the International Health Regulations (IHR) was updated, giving an insight of dynamic of emergency disease management, especially to those who may not be familiar to the timeline of IHR's updates.
- page 2, lines 47-48 (“Due to the unprecedented spread the virus, the world has gone into a virtual lockdown as several countries have initiated strict screening of potential cases introduced in their territory.”): the opening part of this sentence should be changed to “Due to the unprecedented spread of the virus”.
Line 42 , page 1 - previously line 47-48, corrected the missing word of "Due to the unprecedented spread the virus" to "Due to the unprecedented spread of the virus"
- page 2, line 51 (“The authors of this case study will investigate the medical, social, political and economic impact of COVID-19 in India by descriptive observational study from the first case detected in India until the date of submission of the article.”): since the submission date is not reported in this article, it would be helpful to explicitly indicate it or to indicate the time length – possibly expressed in months - of the data considered
Line 46-51 , page 2 - gave more information regarding our research methodology and added specific time frame of study conducted i.e. from first case in 30 January to 12 June 2020.
case presentation
The present section provides an overview of the situation in India, starting from a more general description of India to then focus on India’s health care system. A more in-depth analysis of the case would probably be appropriate.
we have mentioned in the limitation in in line 547-551, page 12, that we conducted a literature overview rather a systematic review, having an in-depth analysis with the limited literature would be difficult. Therefore, we have made an epidemiological overview of the situation of India.
- page 2, lines 72-74 (“Through its federal power division between union and state, governments on a local level have the autonomy to legislate, for example law and public health”): in order to make the sentence clearer and therefore easier to understand, the expression “for example” could be substituted with, for instance, “concerning”.
line 69-70, page 2 - accepted substitution of word "for example" to "concerning"
- page 2, lines 75-76 (“Other religious groups are 75 Muslim (~ 14%), Christian, Sikh, Buddhists, and Jains”): in order to be politically correct, if the percentage for Muslims in India is to be reported, the percentages for the other religious groups should be also added.
line 72-73, page 2 - (Other religious groups are Muslim (~ 14%), Christian (2.3%), Sikh (1.7%), and others/unspecified (2%)) - added percentage of minority religious groups as suggested for fair political view.
other corrections
line 73-74, page 2 - written full form of Sustainable Development Goal (SDG)
line 91-93, page 2 - ("Primary-level health centre includes sub-centres for marginalised populations, rural as well as urban regions for disease prevention and health promotion") - rephrased to make sentence concise and coherent.
- page 3, line 138 (“ICRM has also validated non-US FDA EUA/CE IVD kits for quick commercial use in India”): the acronym “ICMR” was here misspelt to “ICRM”.
line 135-136, page 3 - ("ICMR has also validated non-US FDA EUA/CE IVD kits for quick commercial use in India") - error of ICMR as ICRM has been amended.
- page 4, lines 152-154 (“On 14 March, WHO [25] provided details of the first two events of death, both patients were of age >65 years and with co-morbidities. According to the press release by the Ministry of Health and Family Welfare 76% of confirmed cases were male”): a formula should be found to connect the first sentence with the second one. Moreover, since this paper aims at in-depth analyzing the cases in India from the first ones to the submission date of the article, the authors could expand on the analysis of the first reported cases.
line 146-161, page 4 - added detailed information regarding the index cases and the steps taken of by the Ministry of Health and Family Welfare (MOHFW) in India.
line 162-169, page 4 - ("The availability of segregated data was spare. However, according to the press release by the Ministry of Health and Family Welfare on 6 April, 76% of confirmed cases were male.") - added more information related to the data availability.
- page 4, line 167 (“The doubling time of case-counts steadily decreased t currently 8 days (9 June 2020).”): the authors mistakenly typed a “t” in this sentence.
line 172-173, page 4 - ("The doubling time of case-counts steadily decreased currently to 8 days") - amended "t" to "to"
management and outcome
- page 5, line 183 (“The case-counts of this scenario are based on the total population of the cities Mumbai, Pune, Kochi and Bengaluru located in current COVID-10 hotspots [28].”): the term “COVID-10” should be substituted with “COVID-19”.
line 187-188, page 5 - ("The case-counts of this scenario are based on the total population of the cities Mumbai, Pune, Kochi and Bengaluru located in current COVID-19 hotspots) - amended error of "COVID-10" to "COVID-19"
- page 5, line 205 (“However, an augmented transmissibility was modelled to assess the effect of panic shopping or mass travel prior the implementation of the lockdown could mitigate its effect [30].”): this sentence is articulated in such a way that it is difficult to understand, it could therefore be expressed differently, as to make it easier and more immediate to understand.
line 210-212, page 5 - ("However, an augmented transmissibility was modelled to assess the effect of panic shopping or mass travel prior the implementation of the lockdown. These enhanced exposures could mitigate effect of a lockdown") - additional sentence has been added to give clarification of the meaning of the phrase.
- page 5, line 208 (“Chatterjee, K et al. [31]”): the initial of the first author’s name (“K”) should be omitted as – according to the journal guidelines - references in the text should only include surnames.
line 213, page 5 - The initial of author "K" for Chatterjee, K et al. has been left as it is since there are 2 authors in this research with the same surname and same publication date.
- page 6, lines 237-239 (“Through rigorous point of entry surveillance efforts, by the end of March, India temperature screened more than 1.5 million passengers at the airports, placing thousands of passengers under surveillance in home isolation.”): the expression “temperature screened” is not immediate to understand and could thus be substituted with a another, possibly clearer, expression.
line 242-244, page 6 - ("Through rigorous point of entry surveillance efforts, by the end of March, India conducted thermal scan on more than 1.5 million passengers at the airports, placing thousands of passengers under surveillance in home isolation" ) sentence has been rephrased from "temperature screened" to "conducted thermal scan" for professional usage and clearer understanding.
Issue with reference
line 275-278, page 6 - ("Furthermore, the deviation of modelled projections and the current development of the outbreak might be explained by the low test capacity and therefore, the Indian Government has been criticized for reporting suspiciously suppressed data") - suspicious data has been rephrased as suspiciously suppressed data, and citation has been replaced with a journal than a tweet.
- page 7, lines 280-281 (“Entry screening expanded for other nations as the virus began spreading globally, in line with the International Health Regulations 2005 for point of entry screening”): the date “2005” should be removed.
line 285-286, page 5 - ("Entry screening expanded for other nations as the virus began spreading globally, in line with the International Health Regulations 2005 for point of entry screening") - 2005 has been left stating the reason in line 39-41, page 1.
- page 7, line 285 (“Despite against the international health regulations 2005 [44]”): the sentence seems to be built in an incorrect way, the authors are therefore suggested to rephrase it.
line 290-292, page 7 - ("Although the International Health Regulations advices against reduced mobility in terms of travel") - sentence rephrased from "despite against the international health regulations 2005" to "Although the International Health Regulations advices against reduced mobility in terms of travel" for clearer expression.
-page 7, lines 292-294 (“Passengers’ left hand was stamped with inedible ink to maintain the date and time for home quarantine, a move that could risk assault due to stigma towards COVID-19 suspects (Human Rights Watch, 2020)”): the reported reference does not follow the journal guidelines, stating that references should be numbered in order of appearance and “In the text, reference numbers should be placed in square brackets [ ]”).
line 297-299, page 7 - ("Passengers’ left hand was stamped with inedible ink to maintain the date and time for home quarantine, a move that could risk assault due to stigma towards COVID-19 suspects[50]") - citation error has been amended
- page 8, line 318 (“3.4. Health promotion”): a possible substituting title, which would seem more appropriate for the content of the present section, could be “health recommendations” or, alternatively, “health guidelines”.
line 324, page 8 - title amended to "health recommendations" from "health promotion" as the section talks about methods to maintain good practices and guidelines to mitigate spread of infection.
- page 8, lines 336-343 (“After the imposition of the first lockdown at the end of March, the government released several guidelines on protection methods such as making face masks compulsory in public places, but preferably using reusable mask, leaving medical masks for health professionals. Public etiquettes such as social distancing, avoiding spitting in public and avoiding mass gathering was enforced. Myths surrounding coronavirus arose regarding false remedies, creating panic among the citizens. Following mass misinformation spread among the citizens through social media, the government of India launched a ‘MyGov Corona Helpdesk’ on the highly popular social media application WhatsApp, to receive Indian-centric accurate and verified information.”): no references were reported here; the authors are suggested to add some references, thus allowing the reader to possibly expand their knowledge by consulting the reported sources
line 342-351, page 8 - ("After the imposition of the first lockdown at the end of March, the government released several guidelines on protection methods such as making face masks compulsory in public places, but preferably using reusable mask, leaving medical masks for health professionals [57]. Public guidelines such as social distancing, avoiding spitting in public and avoiding mass gathering was enforced. Following mass misinformation spread among the citizens through social media, regarding false remedies and fake news, the government of India launched a ‘MyGov Corona Helpdesk’ on the highly popular social media application WhatsApp, to receive Indian-centric accurate and verified information [57]") - references have been added to specify finding of information.
- page 8, line 338 (“Public etiquettes such as social distancing, avoiding spitting in public and avoiding mass gathering was enforced.”): the expression “public etiquette” could here be substituted with a more appropriate one as, for instance, “health recommendations”.
line 344-346, page 8 - ("Public guidelines such as social distancing, avoiding spitting in public and avoiding mass gathering was enforced.") - term "etiquettes" replaced with "guidelines" for clearer explanation and proper usage of scientific term.
- page 9, line 368 (“Moreover, Singh et al. [58] expect a downward trajectories of inflation”): the term “trajectories” should be made singular “trajectory”.
line 373-375, page 9 - ("Moreover, Singh et al. [64] expect a downward trajectory of inflation (CPI and WPI) as the oil price falls and productivity and demand decreases") - error of "trajectories" amended to "trajectory"
other correction
line 500-502, page 14 - ("Divided by sector, the projected impact is most for the chemical sector (US$ 129 million), followed by textile and apparel at US$ 64 million and automotive industry at US$ 34 million") - added missed word "million" after US$ 34.
- page 9, lines 386-387 (“it is aimed to help the poorest of the poor [62].”): the expression “the poorest of the poor” is not adequate for the scientific language used in articles and could therefore be changed.
line 392-393, page 9 - ("The volume of the package amounted approximately 1% of the country’s GDP, it is aimed to help the poorest segment of the population") - tense changed from "amounts" to "amounted" as action took place in the past. Sentence rephrased from "poorest of the poor" to "poorest segment of the population" for proper usage of scientific language.
issue with reference
line 407, page 10 - (... challenge” [70] (p. 3). Considering the production capacities of") - reference [70] (p. 3) written as directed by the MDPI journal guideline.
page 10, lines 408-410 (“Newton et al. [66] describes the risks to the supply and quality of tests, drugs and vaccines imposed by insufficient evidence regarding COVID-19 and shortages that thrive substandard or falsified drugs.”): since “Newton et al.” is a plural subject, the verb should also be plural and, therefore, “describe” should be written.
line 414, page 10 - ("Newton et al.[71] describe the risks to the supply and quality of tests, drugs") - amended "describe" to "describes"
conclusion and discussion
Firstly, the title of the present section should be totally rewritten. The the part of the discussion must be extended in an appropriate way, integrating literature that better justifies the results achieved. It should also be clarified in what terms the results presented are original, explaining why for India the situation can be considered particular. The part of the conclusions must be written separately, clearly summarizing the main result obtained from this survey. In addition, the limits and future developments should be highlighted more clearly in the final part.
section rewritten into 2 separate sections, 4. Discussion and 5. Conclusion
4. Discussion
New section added to the research for more coherence of the result of the research conducted.
line 474- 551, page 11 - 12 - section discusses the findings of the paper and relates to the literature overview done and why the case of India has been particular. The section also mentions the limitations of the study.
- page 11, lines 469-470 (“The spread of COVID-19 rooting from China in December 2019 to a global scale has been categorized as a pandemic by the World Health Organization on 11 March 2020”): a specific reference from the World Health Organization could be added here.
line 475-476, page 11 - ("The spread of COVID-19 rooting from China in December 2019 to a global scale has been categorized as a pandemic by the World Health Organization on 11 March 2020") - reference added to this phrase.
page 11, line 472 (“India’s rigorous point-of-entry screening might be of infeasible in order to delay epidemic outbreak.”): this sentence could be re-phrased as to make it clearer and therefore easier to understand
line 478-480, page 11 - ("India’s rigorous point-of-entry screening may be infeasible in delaying the epidemic outbreak as the authorities conducted only thermal scanning for symptomatic passengers, thereby disregarding asymptomatic passengers as disease carriers.") - added phrase for more clarity.
The journal guidelines explicitly state that the article should include “a conclusion briefly outlining the take-home message and the lessons learned”, the authors could therefore clearly define the take-home message for this study.
5. Conclusion
line 552- 588, page 12-13 - conclusion has been separated from the discussion section and now mentions the summary of the findings and take-away points of the study. New phrases are added and previous once are deleted with the track changes function.
- page 11, line 478 (“India’s healthcare system maybe be dented if there is a superfluous hospital admission due to 477 lack to infrastructure and medical experts.”): the expression “lack to” should be changed to “lack of “.
line 562, page 13 - ("admission due to lack of adequate infrastructure and medical experts in relation") "lack to" corrected to "lack of".
references
Concerning the referencing, the authors seem to respect the journal guidelines as for in-text citations (“In the text, reference numbers should be placed in square brackets [ ], and placed before the punctuation; for example [1], [1–3] or [1,3].”). Conversely, the authors do not seem to follow the journal guidelines for references at the end of the text, therefore, major changes should be applied to this section. Moreover, the authors do not include the DOI (digital object identifier) for articles, as explicitly suggested in the aforementioned guidelines.
the references have been amended as the journal guideline suggests and DOI of scientific articles have been added. Also, non-scientific articles were chosen for some information due to lack of scientific publication, as well as alternate concept mentioned in public domain in comparison to data presented by the health authorities.
Round 2
Reviewer 1 Report
Dear authors,
I keep commenting that the article is not an observational study. This is a review article. Also, the text still contains a high percentage of plagiarism.
Author Response
Thank you for the feedback.
I keep commenting that the article is not an observational study. This is a review article
We have made edits in the methodology, please see
Line 46-48, Page 2 - The study is a literature overview and part of a case series on various countries’ initial experiences on the COVID-19 pandemic.
Also, the text still contains a high percentage of plagiarism.
Our research mentor has provided us the certificate from the internal system that checks for plagiarism (I attach a certificate). There are less than 5% of inclusions, since we use data from open sources, the work contains citations, but they are all accompanied by links.
The attachment is a combination of the certificate of plagiarism and the case study.

Reviewer 2 Report
Dear Authors,
the article is really improved.
Sincerely
Author Response
Thank you very much for you honest and constructive feedback.
Sincerely